# Multi-omic integration of microbiome data for identifying disease-associated modules

Efrat Muller ®[1], Itamar Shiryan[1] & Elhanan Borenstein ®[1,2,3] ✉

Multi-omic studies of the human gut microbiome are crucial for understanding its role in disease across multiple functional layers. Nevertheless, integrating and analyzing such complex datasets poses significant challenges. Most notably, current analysis methods often yield extensive lists of disease-associated features (e.g., species, pathways, or metabolites), without capturing the multi-layered structure of the data. Here, we address this challenge by introducing "MintTea", an intermediate integration-based approach combining canonical correlation analysis extensions, consensus analysis, and an evaluation protocol. MintTea identifies "disease-associated multi-omic modules", comprising features from multiple omics that shift in concord and that collectively associate with the disease. Applied to diverse cohorts, MintTea captures modules with high predictive power, significant cross-omic correlations, and alignment with known microbiome-disease associations. For example, analyzing samples from a metabolic syndrome study, MintTea identifies a module with serum glutamate- and TCA cycle-related metabolites, along with bacterial species linked to insulin resistance. In another dataset, MintTea identifies a module associated with late-stage colorectal cancer, including *Peptostreptococcus* and *Gemella* species and fecal amino acids, in line with these species' metabolic activity and their coordinated gradual increase with cancer development. This work demonstrates the potential of advanced integration methods in generating systems-level, multifaceted hypotheses underlying microbiome-disease interactions.

The human gut microbiome is an incredibly complex ecosystem that has a marked, multifaceted impact on our health[1–3], and a well-established role in the pathogenesis of numerous diseases[4–8]. This appreciation for the importance and complexity of the gut microbiome has prompted a proliferation of multi-omic microbiome studies, which apply several molecular assays to the same set of samples, in the hope of capturing multiple layers of information concerning the involvement of the microbiome in disease. Among such multi-omic studies, one increasingly popular study design, for example, relies on collecting paired high-throughput microbiome and metabolome profiles[9–13] (e.g., coupling shotgun metagenomics sequencing and

mass spectrometry). Unfortunately, however, while such multi-omic data clearly offer an exciting opportunity to study the microbiome and its role in human health, rigorous integrative analysis of such data remains highly challenging, as does using such data to gain a systems-level understanding of the microbiome[14–16].

A common aim of microbiome studies, and accordingly a key goal of many multi-omic microbiome analyses, is to identify disease-associated markers—specific features from the various omics (e.g., certain species, pathways, or metabolites) whose measured abundances are strongly associated with the disease in question[17]. Such markers can later be used to predict disease, informing, for example, microbiome-based diagnosis[18–20], or to suggest novel hypotheses

[1]Blavatnik School of Computer Science, Tel Aviv University, Tel Aviv, Israel. [2]Faculty of Medical and Health Sciences, Tel Aviv University, Tel Aviv, Israel. [3]Santa Fe Institute, Santa Fe, NM, USA. ✉e-mail: elbo@tauex.tau.ac.il

concerning specific microbiome components or mechanisms that are at play, guiding future experimental and clinical studies. Given disease and control samples (or, more generally, some phenotypic measure of interest), disease-associated markers can be identified using a variety of statistical approaches, ranging from univariate statistical tests that are applied independently to each feature from each omic, to multivariate statistical methods that consider potential statistical dependencies between all features, such as partial least squares (PLS) or linear regression[10,21–24]. More recently, machine learning (ML) methods have also been applied widely for this purpose by first training some ML model for predicting the phenotype based on all features (either from a single omic or from multiple omics combined), and then using various model explainability methods for detecting informative features in this model[11,12,25–28]. Ultimately, however, such analyses generally result in long lists of features (from one or more omics) that are associated with the disease, without leveraging the multi-layered structure of multi-omic data and without offering clear, interpretable hypotheses concerning specific and coherent mechanisms underlying microbiome-disease interactions.

Several preliminary attempts directed at addressing this challenge and considering both cross-omic dependencies and associations with a disease state have been introduced. For example, some microbiome studies took a two-step approach by first identifying features that are associated with the disease, and then clustering these features based on pairwise correlations between features[29]. This approach, however, may fail to identify features that are not sufficiently informative by themselves but can be incorporated into larger modules that, as a whole, are strongly predictive of the disease. Other studies aiming to tackle a similar challenge constructed cross-omic correlation networks while also including phenotypes of interest in the network[30], or alternatively constructed separate networks for each host condition and then explored the differences between the networks to identify condition-specific motifs[31–33]. These approaches, however, are often hard to interpret given the high dimensionality of omic data, and the massive correlation networks that are accordingly produced.

A particularly intriguing approach that may be utilized for addressing such analyses originates from the domain of multi-view learning, and is known as "intermediate integration"[34,35]. Unlike traditional multi-view approaches that directly combine raw features, either by naïvely concatenating different omics into a single joint table (i.e., "early integration") or by modeling each omic independently and then creating a final ensemble model (i.e., "late integration"), intermediate integration seeks to combine features from the various omics (or 'views') into an intermediary representation level before utilizing them for downstream tasks such as classification[34] (Supplementary Fig. S1). This approach thus captures dependencies between omics and could accordingly be beneficial for generating multifaceted biological hypotheses. Canonical correlation analysis (CCA), for example, is a popular intermediate integration method that receives two feature tables, and outputs a linear transformation per table so that the resulting latent variables are maximally correlated[36]. CCA extensions that may be particularly relevant for microbiome multi-omic data include sparse CCA (sCCA)—a CCA extension that includes sparsity constraints to deal with large numbers of features, and sparse generalized CCA (sGCCA), which generalizes sCCA to further support more than two views[36,37]. Indeed, CCA and its extensions have been previously applied to microbiome-related multi-omic data with insightful results[38–43]. One recent study, for example, applied sCCA to microbiome taxonomic profiles and host transcriptomics data from patients with either irritable bowel syndrome, inflammatory bowel disease (IBD), or colorectal cancer (CRC), identifying host-microbiome associations that are shared across these cohorts and others that are disease-specific[38]. In another study, sCCA was used to explore the association between early life gut microbiome and metabolome,

finding that the specific taxa and metabolites driving these associations at 6 weeks of age differed from those at the age of 12 months[40]. Importantly, the CCA transformation can further take the disease state of each sample into account, thus seeking representations that highlight interactions both across different omics and between these omics and the disease[36,44,45]. Galié et al.[44], for example, studied the effects of a Mediterranean diet on circulating metabolites and used an extension of sGCCA to identify signatures of dietary intervention reflected in both the gut microbiome and plasma metabolites in concert[46]. These studies testify to the potential benefit of such CCA-based models, yet, their applicability to diverse microbiome datasets, and the robustness of the obtained results, remain unclear. Moreover, CCA and its extensions, as well as similar multivariate linear statistical methods, are generally highly sensitive to small perturbations in the data, parameter choices (e.g., sparsity constraint), atypical samples, and collinearity between variables[47–50], requiring careful attention and interpretation when applied to complex microbiome data.

Setting out to address the challenges above and to allow researchers to gain systems-level insights into coherent mechanisms underlying microbiome-disease interactions, here, we introduce a comprehensive intermediate integration-based method (combining CCA extension, consensus analysis, and a validation protocol), which we term "MintTea", for analyzing multi-omic microbiome data. We hypothesize that each such mechanism may involve various taxa, functions, and metabolites which act in concert in disease, and thus MintTea aims to identify robust "disease-associated multi-omic modules", each comprising a set of features from the various omics that both exhibit coordinated variation across omics and, as a whole, associate with the disease or phenotype of interest. We applied MintTea to 9 diverse case-control cohorts with available shotgun metagenomics data (processed into both taxonomic and functional profiles), 6 of which further included fecal or serum metabolomics data. We demonstrated that MintTea was indeed able to capture modules that had high predictive power of the disease (often comparable with that achieved by using all features), while also exhibiting significant correlations between features from different omics. We also showed that some of the modules identified by MintTea recapitulated previous observations concerning the role of the gut microbiome in disease, and provided a catalog of identified multi-omic modules across all datasets analyzed. This work thus serves as a proof of concept for the benefit of advanced integration methods in generating integrated multi-omic biological hypotheses underlying microbiome-disease associations.

## Results

### Introducing MintTea: a framework for identifying robust disease-associated microbiome multi-omic modules

Inspired by the potential of intermediate integration methods described above, we developed a multi-omic integration framework, termed "MintTea" (Multi-omic INTegration Tool for microbiomE Analysis), for identifying sets of features from multiple different omics that are strongly associated both with the disease and with each other (Fig. 1A). MintTea is based on sparse generalized canonical correlation analysis (sGCCA) and other previously introduced related methods[36,44,51,52], and further applies repeated sampling, consensus analysis, and module evaluation to account for noisy data and to ensure robust and confident results.

Briefly, MintTea receives two or more feature tables describing different omics obtained for the same set of samples, as well as a label (e.g., healthy vs. disease) for each sample, as input. Following filtration of rare features and other preprocessing steps (see Methods), MintTea encodes the label as an additional omic (containing a single feature) as previously suggested[44,51,52], and then applies sGCCA, searching for a sparse linear transformation per feature table that yields maximal correlations between the respective latent variables, as well as between

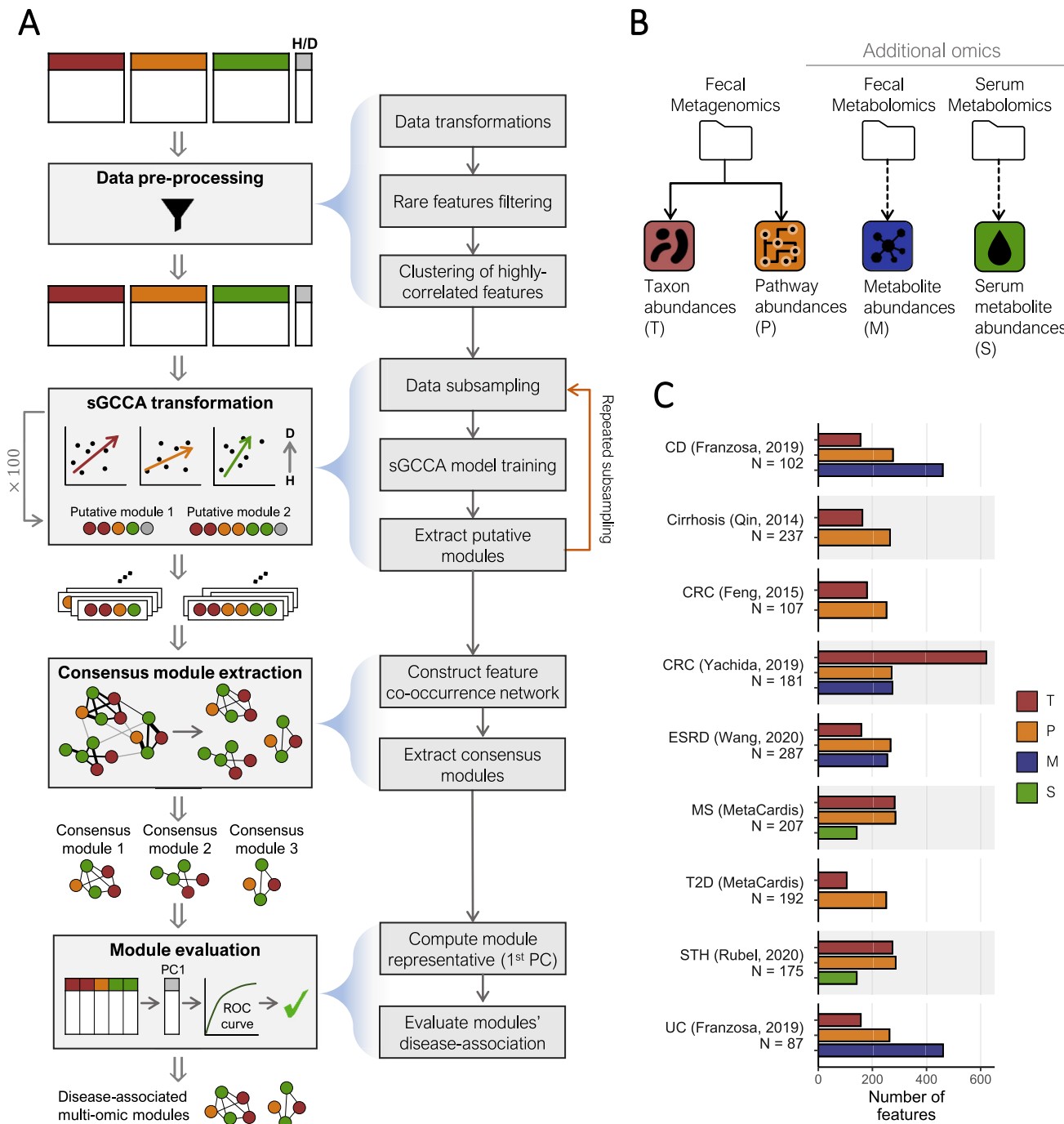

**Fig. 1 | MintTea pipeline illustration and the multi-omic datasets analyzed in this study. A** An illustration of the MintTea pipeline, including data preprocessing, repeated module discovery using sparse generalized CCA (sGCCA), consensus analysis, and evaluation of each module's association with the disease. The letters "H" and "D" represent the terms "healthy" and "disease", respectively. See Methods. **B** Data types used in this analysis. Throughout the manuscript, taxonomic, pathway, fecal metabolite, and serum metabolite features are labeled with the letters 'T', 'P', 'M' and 'S', respectively. **C** The number of features available from each omic in each dataset after preprocessing. Sample sizes are noted under dataset names. CD crohn's disease, CRC colorectal cancer, ESRD end-stage renal disease MS metabolic syndrome, T2D type-2 diabetes, UC ulcerative colitis.

these variables and the label (Fig. 1A top). This procedure yields a latent variable per feature table (omic), which is a sparse linear combination of the various features. We then record the set of features that were assigned non-zero coefficients across the various omics, and define this set as the first "putative module". sGCCA can then find additional sets of latent variables that are orthogonal to the previous ones, by deflation on those previous latent variables, with each iteration similarly providing a new putative module. Next, to identify specific modules of features that are robust to small changes in the input

data, MintTea repeats the entire process multiple times on random data subsets (e.g., 90% of the samples) and records the resulting putative modules from each such iteration. It then constructs a co-occurrence network, where features are connected if they consistently co-occurred in the same putative module (e.g., over 80% of total iterations), and identifies "consensus modules" (i.e., connected subgraphs; see Methods; Fig. 1A middle). Importantly, as MintTea balances between two objectives, namely associations between omics and associations with the phenotype, the resulting consensus module may

capture only one of these two objectives. Accordingly, to output robust, disease-associated, and cross-omic correlated modules, MintTea finally evaluates each consensus module, discarding modules that do not adhere to the desired criteria. To this end, it first filters out any module in which the average cross-omic correlation is not higher than that obtained for random modules. Then, MintTea applies principal component analysis (PCA) to all the features included in each remaining module, uses the first principal component (PC) of each module as the module's representative, quantifies how well this one representative PC predicts the disease using the area under the receiver operating characteristic curve (AUC), and preserves only modules for which this AUC is >0.7 and higher than AUC obtained for random module (of the same size and distribution over omics; see Methods; Fig. 1A bottom). The final outcome of MintTea is therefore multiple "disease-associated, multi-omic modules", each capturing features from multiple omics that are highly correlated across the different omics, and also with the disease.

In Supplementary Note 1, we discuss the conceptual differences between MintTea and other related approaches, and quantitatively compare MintTea to other sGCCA-based methods[36,44], demonstrating that MintTea provides a good balance across multiple desired properties, while achieving a substantially lower false discovery rate.

## Using MintTea for identifying multi-omic modules across cohorts and diseases

To examine MintTea's ability to identify disease-associated multi-omic modules in various settings, we obtained 9 different datasets of shotgun metagenomics sequencing. All datasets are from case-control studies, spanning multiple health conditions and diseases, including Crohn's disease (CD), ulcerative colitis (UC), CRC, metabolic syndrome (MS), type-2 diabetes (T2D) and others. For each dataset, we generated 2–3 different feature tables ('views') based on the available omic data. Specifically, in all datasets, metagenomic data was available and processed into taxonomic profiles ("T") and pathway-level functional profiles ("P") (see Methods). In addition, for 3 of the datasets, fecal metabolomics data ("M") were also available, and for 2 others datasets, fasting serum metabolomics data ("S") were available. Figure 1B, C summarizes the omics and datasets used for the main analysis (and see also Supplementary Data File S1).

Applying MintTea to these 9 datasets, we found 2–5 consensus modules per dataset, each comprising 2 to 38 features from the various omics (average: 13.23; Fig. 2A; Supplementary Data File S2; Supplementary Data File S3). Most of these modules captured correlations between features from different omics that significantly exceed those of randomly sampled modules (Fig. 2B), with 1–2 modules per dataset also exhibiting strong association with the disease, as defined above (Fig. 2C; Supplementary Data Files S2 and S4). In the T2D (MetaCardis[13,53]) dataset, for example, we identified 5 consensus modules, 2 of which also passed our evaluation and were classified as disease-associated multi-omic modules, with AUCs reaching 0.86 (based on the 1st PC of the 21 features in this module) and 0.73 (based on only 2 features in this module), compared to 0.71 and 0.64 on average, respectively, when evaluating randomly selected modules. In the late-stage CRC dataset (Yachida, 2019[11]), as another example, we found 2 consensus modules, one of which was classified as a disease-associated multi-omic module with an AUC of 0.72 compared to only 0.63 using equivalent, randomly selected modules (more on this module below). The entire list of identified modules, as well as detailed statistics about their association with disease and correlations between features across omics, is provided in Supplementary Data Files S2–S4).

While most of our analyses below focus only on disease-associated multi-omic modules (i.e., modules that passed our evaluation phase and adhere to the criteria we defined), we also sought to evaluate the overall predictive power of all identified consensus modules in each dataset in order to gain a better understanding of

CCA-based methods' ability to identify informative axes of variation in the data. To this end, we trained random forest (RF) models on the 1st PCs of all modules in each dataset, and estimated their combined predictive accuracy (with cross-validation). We further compared the AUC's of these models to RF models trained on the entire concatenated multi-omic data (referred to as an "early integration" approach; see Methods; Supplementary Data File S5). We found that in most cases, these consensus modules by themselves achieved an AUC comparable to that of the early-integration models, while utilizing a substantially smaller number of features in total (Supplementary Fig. S2). In the UC (Franzosa, 2019[12]) dataset, for example, the 4 consensus modules identified by MintTea (totaling 73 features) achieved a total AUC of 0.885 (SD: 0.12), where each module is represented by its 1st PC as explained above. In comparison, the early integration approaches achieved a similar AUC of 0.882 (SD: 0.14), but used 235 features on average for training the RF model *after* feature selection. Similarly, the overall AUC of the CRC (Feng, 2015)[54] modules was 0.839 (SD: 0.15), utilizing 36 features in just 3 modules (and hence based on just 3 values), compared to 0.824 (SD: 0.15) using early-integration-based on 103 features.

It should also be noted that the features included in the disease-associated multi-omic modules reported above, did not necessarily overlap with those ranked highest by the early integrated multi-omic models. Specifically, on average, only 63.9% (SD: 24.8%) of the features comprising each disease-associated module (as defined above), were also significant contributors to the multi-omic early-integration models (see Methods, Supplementary Data File S6). This observation may attest to the ability of the intermediate integration approach to consider features in the context of their associations with other features and not only their association with disease.

## Disease-associated multi-omic modules reveal multifaceted biological signatures

Figure 2D presents one example of a MintTea module identified in the late-stage CRC dataset[11]. This module included 7 features in total; 3 bacterial species, namely *Peptostreptococcus stomatis*, *Peptostreptococcus anaerobius*, and *Gemella morbillorum*, and 4 fecal metabolites: 2 branched-chain amino acids (BCAA's), namely valine (Val) and leucine (Leu), an aromatic amino acid - phenylalanine (Phe), and cysteine-glutathione disulfide divalent. The 1st PC of the 7 features included in this module, by itself, yielded an AUC of 0.72 in classifying disease state (Fig. 2C), and the average Spearman correlation between features from different omics was 0.28 (all correlations had an FDR-corrected *p* value < 0.05, Supplementary Data File S4). All features in this module, except for the last, were significantly elevated in disease (Mann–Whitney tests, FDR < 0.05), and were also reported as disease biomarkers in multiple other independent studies[55–59]. Specifically, in the original paper by Yachida et al[11]., all three species were overrepresented consistently across all disease stages, as opposed to other species that were elevated in specific stages only. A similar trend was reported for the three amino acids (Val, Leu and Phe), indicating a coordinated pattern with the above taxa[11]. *P. stomatis* and *G. morbillorum* were also found to have significantly high replication rates compared to controls across all cancer stages in that study, hinting at their increased metabolic activity during cancer development. One possible explanation for *Peptostreptococcus* species appearing together with the BCAA's is their involvement in BCAA metabolism. Interestingly, *Peptostreptococci* species were specifically identified as major fermenters of Phe and Leu, while the most prominent CRC-associated species, *Fusobacterium nucleatum*, for comparison, was reported to prefer different amino acid substrates in the same review study[60]. Phenylalanine was also significantly and positively correlated with both *P. stomatis* and *G. morbillorum* in a recent meta-analysis[61]. Lastly, though we could not recover a mechanistic link between cysteine-glutathione disulfide and the other features, we note that the

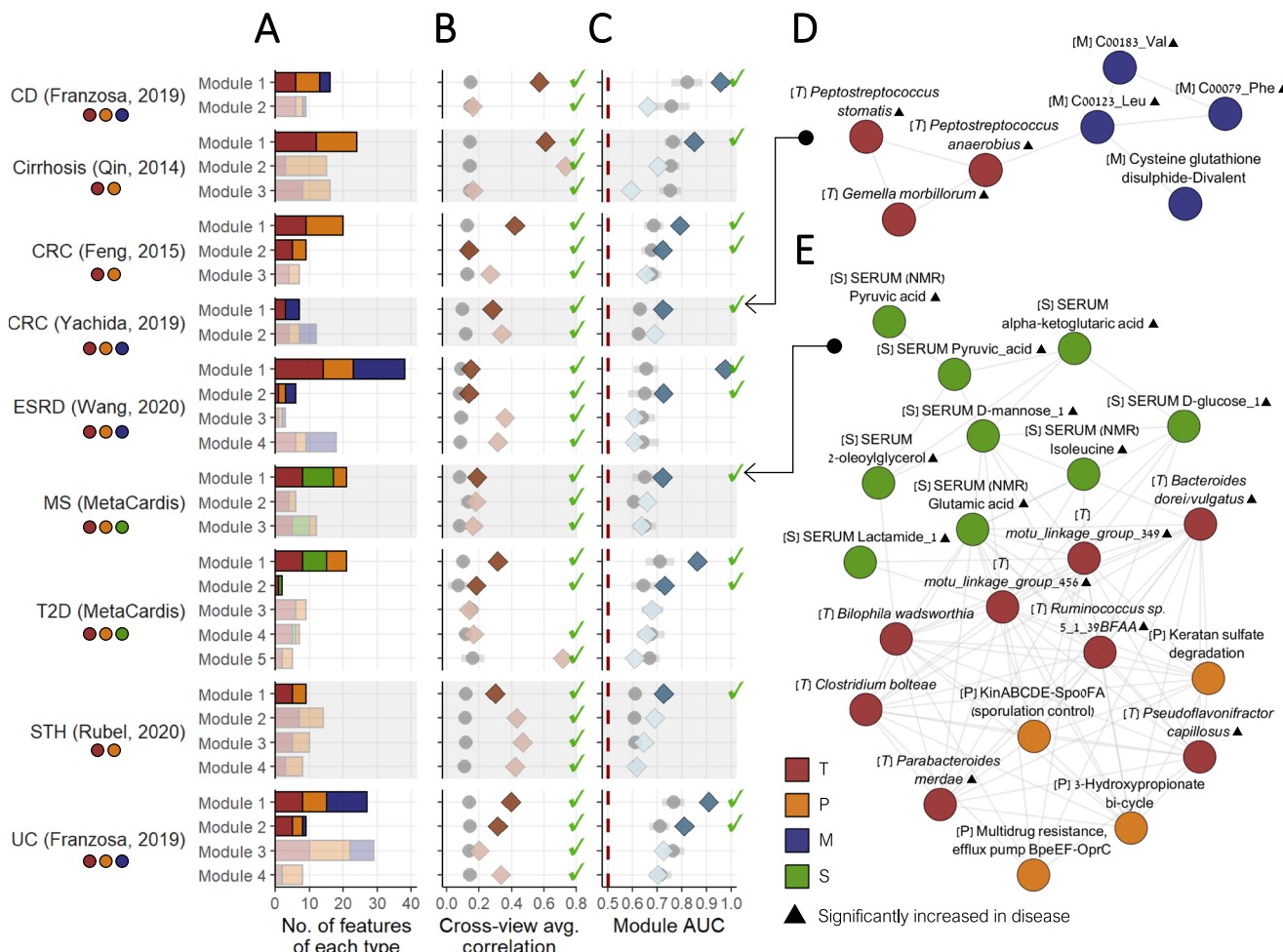

**Fig. 2 | Identifying disease-associated multi-omic modules via MintTea intermediate integration framework.** A–C Properties of the multi-omic consensus modules obtained by applying MintTea to several key datasets, including **A** the number of features in each module, stratified by feature type, **B** cross-omic correlation per module, calculated as the average pairwise Spearman correlation between features of different omics (with gray points and lines indicating the average and standard deviation of correlations obtained from random modules of the same size), and **C** AUC per module, calculated using the first principal component values against disease label of each sample (with gray points and lines indicating the average and standard deviation of correlations obtained from random modules, as before). Modules that only included features from a single omic were discarded from our analyses but are listed in Supplementary Data Files S2 and S3. Modules that exhibit between-omic correlations (higher than random modules) and that are disease-associated (AUC > 0.7 and above that of random modules) are shown in a darker color. Circle colors under each dataset name indicate which omics were available for this dataset. The sample size of each dataset (i.e., the number of individuals profiled) are as shown in Fig. 1C. **D** A late-stage CRC-associated multi-omic module. Node colors represent feature types (see Fig. 1B). Edges connect features that appeared together in an sGCCA putative module in >80% of data subsampling iterations. Correlations between pairs of features within the module can be viewed in Supplementary Data File S4. **E** An MS-associated multi-omic module.

correlations between this metabolite and the other 3 amino acids were strikingly high, even when controlling for disease state (with all Spearman and partial Spearman correlations $\rho > 0.47$ and FDR $< 1 \cdot 10^{-10}$).

Another example, illustrated in Fig. 2E, is a module identified in the metabolic syndrome (MS) cohort of MetaCardis (these are subjects with MS according to International Diabetes Federation criteria, without type 2 diabetes (T2D) or coronary artery disease). This module included 8 species (2 of which lacking taxonomic annotation), 4 pathways, and 9 serum metabolites (measured by either NMR or mass spectrometry), with the 1st PC of these features achieving an AUC of 0.72 for predicting MS. Notably, the interplay between the gut microbiome and serum metabolites is significantly less straightforward compared to that observed with fecal metabolites, as it is likely overshadowed by other physiological processes and systemic metabolic activities[22,62,63]. Indeed, the average correlation between features from different omics in this model was lower than those in modules of other datasets, but nonetheless higher than randomly sampled

modules (Fig. 2B). Specifically, 19 of 108 correlations between serum metabolites and gut microbiome features in this module were significantly positive (FDR < 0.05; for comparison, in random modules of the same size, less than 1 metabolite-microbiome feature pair was significantly correlated). Of the serum metabolites included in this module, several are well-known signatures of metabolic disorders, including increased glucose, mannose[64], TCA cycle metabolites (e.g. pyruvic acid and alpha-ketoglutaric acid)[65], and glutamate[66]. Circulating levels of isoleucine (as well as other BCAA's and BCAA-related metabolites) have also been repeatedly shown to play a role in glucose homeostasis and metabolic syndrome risk[65,67,68]. Though some of the other module's metabolites have not been specifically discussed in the context of metabolic disorders, they are all significantly positively correlated with one another (32 of 36 metabolite pairs are positively correlated with an FDR < 0.05), effectively representing a cohesive cluster of metabolites coordinately elevated in MS. Interestingly, in a recent study, several of these metabolites (pyruvic acid, glucose, glutamic acid) were found to be significantly well-predicted by the gut

microbiome when measured in feces, but only marginally well-predicted in blood[63], further strengthening the assumption that the gut microbiome's association with these serum metabolites is likely obscured by additional processes, but nonetheless existent.

Moreover, the inclusion of certain bacterial species in this module along with these metabolites is likely related to their previously reported role in glutamate metabolism and/or BCAA metabolism. *Bacteroides dorei/vulgatus*, for example, though only marginally significantly increased in MS patients in our dataset (FDR = 0.07), have been shown to impact serum isoleucine levels[69]. *B. vulgatus* specifically was also reported to take part in glutamate metabolism[70] and further recognized as one of the main species driving the association between biosynthesis of BCAAs in the gut and insulin resistance[7,71]. The exact role of *B. vulgatus* in the development of metabolic disorders, however, remains unclear and is probably context- and strain-dependent, as it has been shown to confer health benefits (including metabolic-related) in some studies, while positively correlating with insulin resistance and body fat in others[7,69,72–75]. Interestingly, *B. vulgatus* was also reported to induce compositional shifts when administered to rats, specifically promoting an increase in *Parabacteroides* species that is also included in this module[72]. *Parabacteroides mardeae*, another species in this module, has also been shown to enhance BCAA catabolism, again with positive health benefits[76], albeit significantly increased in MS patients in our dataset. Lastly, we note that *Bilophila wadsworthia* has been mostly discussed in the literature in the context of IBD and colon inflammation[77], but has also been implicated in metabolic dysfunctions[78]. It was also found to cluster with *Clostridium bolteae*, another species in the module, in terms of their responses to short and long-term diet[79], possibly explaining their co-occurrence in this module. As for the pathway features included in this module, likely not all represent specific metabolic functions relevant to MS, but nonetheless exhibit strong correlations with some of the modules' species, suggesting potential dependencies. Overall, we find multiple reports supporting specific links within the module, but the shared role of these bacterial species in MS phenotypes and circulating levels of MS-related metabolites specifically remains elusive.

Importantly, even when analyzing metagenomic-derived features alone (i.e., in the absence of additional omics), MintTea may highlight interesting insights related to associated taxonomic and functional disease biomarkers. In the liver cirrhosis dataset[32], for example, we found a module (Supplementary Fig. S3A) that included multiple species typically found in the human oral cavity, including several *Streptococcus*, *Veillonella*, and *Megasphaera* species (both commensal and opportunistic pathogens)[80], all of which previously found to be increased in stool samples of individuals with liver diseases[81–83], as well as other gastrointestinal disorders[84–86]. The co-occurrence of multiple *Streptococcus* and *Veillonella* species in this module is in line with their strong co-abundance patterns, confirmed across multiple diverse large-scale cohorts[87], and also with known metabolic interactions between them[88]. Furthermore, several pathways in the module relate to vitamin K metabolism (menaquinol-8 biosynthesis I, menaquinol-11 biosynthesis, 2-carboxy-1,4-naphthoquinol biosynthesis, phylloquinol biosynthesis, geranylgeranyl diphosphate biosynthesis), also relevant to liver health[89,90]. Indeed, several vitamin K forms (including menaquinone) are known to be bacterially produced in the human intestines, with *Veillonella parvula* specifically reported to produce menaquinones in vitro[91]. In another study, menaquinone biosynthesis was found to be associated with *Streptococcus* overgrowth in IBD[87]. Additional pathways in the module, e.g. those related to L-methionine and heme b biosynthesis, are also possibly tied to menaquinone metabolism and to *Veillonella*'s specific metabolic activities. L-methionine's role, for example, as a methyl group donor for certain menaquinones biosynthesis has been confirmed in several bacterial species[92], though not specifically in the species of this module. Also, previous genomic

analysis of *Veillonella* species showed that they harbour complete gene sets for heme biosynthesis[93]. Overall, this module suggests a potential multi-layer vitamin k-related and oral bacteria-driven mechanism that is highly associated with liver damage. Supplementary Fig. S3 further illustrates two additional multi-omic modules associated with IBD, as additional examples.

## Multi-omic modules across diverse cohorts

Having shown that MintTea can identify multi-omic modules exhibiting associations both across omics and with a disease, we finally sought to detect prevalent and recurring multi-omic modules shared across diverse datasets. Due to the substantial technical and methodological differences between the multi-omic datasets analyzed, which render cross-study comparisons impractical, here we focused on shotgun data only, where both taxonomic and functional (pathway-level) profiles were generated uniformly by the curatedMetagneomicData resource[94]. Specifically, we utilized the 3 datasets, previously presented, from the curatedMetagonmicData resource (i.e., the CRC[54], Cirrhosis[95], and STH[96] datasets; Fig. 1C), as well as 11 additional datasets from the same resource (Supplementary Data File S1; Methods), totaling 14 uniformly processed datasets. After applying MintTea to these datasets (Supplementary Fig. S4; Supplementary Data Files S3 and S4), we quantified the overlaps between modules from the different datasets and disease states, and identified significant overlaps (Fig. 3A; Supplementary Data File S7; Methods). Of the 43 consensus module pairs with an overlap of at least 2 features, 10 exhibited a statistically significant overlap (Fisher's exact test, FDR-corrected $p$ value < 0.1), suggesting prevalent multi-omic associations between microbiome features.

We additionally identified specific bacterial species that appeared in multiple modules of different datasets. *Bacteroides uniformis*, for example—a gram-negative, fiber-degrading, highly abundant commensal in the human gastrointestinal tract[97–99]—appeared in modules of 5 different datasets (Fig. 3B). Other species commonly appearing with *B. uniformis* in MintTea modules were also from the *Bacteroides* genus, including *B. xylanivsolvens*, *B. ovatus*, and *B. thetaiotaomicron*. Interestingly, *B. uniformis* tended to appear with vitamin B-related pathways such as pyridoxal 5'-phosphate biosynthesis pathways (the active form of vitamin B6) and thiamine (vitamin B1) diphosphate synthesis. Indeed, *B. uniformis* genomes, as well as the other Bacteroides genomes, are known to have a relatively high coverage of biosynthetic pathways for multiple vitamins, including B-vitamins[100,101]. Furthermore, in a previous study[102], we found that the *Bacteroides* genus is significantly and consistently correlated with fecal Pyridoxamine levels (a form of vitamin B6), based on a meta-analysis of 5 different datasets in which both this genus and this metabolite were detected. Though this analysis was done at the genus level only, we also know that *B. uniformis*, *B. thetaiotaomicron*, and *B.ovatus* (all of which shown in Fig. 3C) are among the most prevalent *Bacteroides* species in adults[103]. Lastly, examining the taxon-stratified outputs of HUMAnN3, we observed that *B. uniformis* was one of the top contributors of reads assigned to these vitamin B-related pathways, further explaining their co-occurrence in the same modules. *B. uniformis* was also a top contributor to other pathways that co-appeared with it across modules from diverse datasets, including 6-hydroxymethyl-dihydropterin diphosphate biosynthesis I and lipid IV$_A$ biosynthesis (which is part of the biosynthesis process of lipopolysaccharide (LPS), a crucial component of bacterial outer membrane). Taken together, these modules suggest a common axis of variation shared between multiple cohorts and characterized mainly by *Bacteroides* species and metabolic processes essential for cell survival, including biosynthesis of several cofactors and LPS.

Noticing the tendency described above of species from the *Bacteroides* genus to co-occur in modules, we conducted a permutation-based analysis to test whether species in MintTea modules are indeed

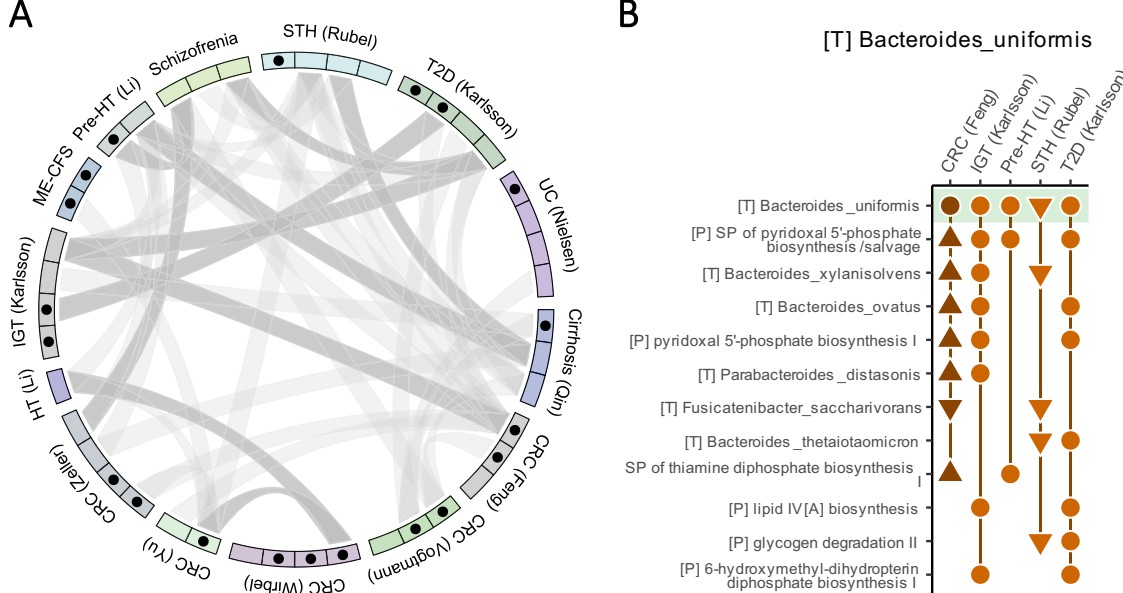

**Fig. 3 | Multi-omic modules across datasets. A** Overlaps between multi-omic modules of different datasets. Each sector in the circus plot represents a consensus module, grouped by the datasets to which they belong. Black dots represent disease-associated multi-omic modules (as previously defined). Links between modules indicate overlaps of at least 2 features, with darker links indicating statistically significant dependences (Fisher's exact test FDR < 0.1). **B** Multi-omic modules from multiple datasets that include *B. uniformis*, and additional overlapping features. For each module (from the dataset listed on top), all module features that appear in at least one other module are presented. Triangles pointing up indicate that the feature level was significantly increased in disease in that dataset, while triangles pointing down indicate an opposite trend (Mann–Whitney tests, FDR < 0.1). Circles indicate no significant difference between study groups. The module in dark orange (CRC, Feng) was also associated with disease state. STH soil-transmitted helminths, HT hypertension, ME-CFS myalgic encephalomyelitis/ chronic fatigue syndrome, IGT impaired glucose tolerance, SP super-pathway. Also see abbreviations in the legend of Fig. 1.

significantly more likely to appear with other species from their respective genus, compared to random modules. We confirmed this was indeed a general trend (22.8% compared to 12.7% [7.7–18.6%], *p* value: 0.001). This observation is in line with the hypothesis that phylogenetically related species are more likely to compete for a shared niche, and, accordingly present stronger co-occurrence patterns.

## Discussion

In this work, we set out to enhance the analysis of microbiome multi-omic data, allowing researchers to identify robust, disease-associated microbiome signatures across multiple molecular processes and to generate multifaceted hypotheses concerning the role of the microbiome in disease. We specifically utilized an "intermediate integration" approach to analyze multi-view (multi-omic) human gut microbiome data, presenting a framework (termed "MintTea") for simultaneously identifying sets of features from different omics that are both highly associated with one another and jointly associated with the studied disease. Our "views" included taxonomic and pathway functional profiles derived from shotgun metagenomics data, as well as metabolomic profiles (from either feces or serum). Applied to multiple diverse case-control datasets, MintTea was able to pinpoint cohesive multi-omic modules of features that reflected coordinated shifts in the data across omics, while also being highly predictive of disease.

Using MintTea, we identified, for example, a CRC-associated module driven by *Peptostreptococcus* and *Gemella* species and the metabolism of specific amino acids. In another cohort of patients with metabolic syndrome, we found a module that included a highly correlated cluster of fasting serum metabolites (mostly glutamate- and TCA cycle-related metabolites) as well as bacterial species that were previously implicated in insulin resistance/ metabolic disorders as well as in isoleucine metabolism. Applying MintTea to a large collection of shotgun datasets, and comparing

the consensus modules identified in each dataset, we further identified multiple modules with a significantly high overlap, suggesting common axes of variation in the gut's microbiome composition and function, shared across multiple cohorts. Future work could specifically search for shared modules across different cohorts of the same disease, or shared modules spanning additional omics (or clinical data, for example, from electronic medical records), to gain more clinically relevant insights into disease-related microbiome processes. Due to data availability limitations, this proposed analysis remained outside the scope of this current work.

Intermediate integration methods, in general, can serve as an alternative (or complementary) analytical approach to early integration, in which different omics are simply concatenated together prior to the application of some ML model (an approach we used in this study as a baseline of disease predictability). Indeed, intermediate integration methods have been gaining increased attention in recent years, in concordance with the proliferation of multi-omic study designs, due to their ability to uncover intricate multivariable interactions, which is desperately required when dealing with complex biological systems[35]. Our motivation to use a sCCA-based method was the assumption that gut microbiome-disease associations may be manifested in microbiome multi-omic data as one or more "modules" of features from all omics that take part in a specific mechanism effected by or promoting host disease. Furthermore, sCCA-based frameworks have been previously applied to various microbiome multi-omic data with insightful results[38,39,45]. To overcome challenges related to the high-dimensionality of the data and consequent instability of results, however, we further expanded this analytical technique with repeated iterations using data subsamples and consensus analysis, to identify robust consensus modules based on multiple sCCA-based putative modules. This conservative approach indeed resulted in markedly lower risk of false discovery, while maintaining similar performance across other evaluation

metrics, when compared to similar methods (see Supplementary Note 1).

In essence, MintTea is a data-driven hypothesis generation tool that aims to discover microbiome signatures reflected by multiple layers of information. The evaluation of such signatures, however, is nontrivial, and remains an open challenge and a major limitation of multiple integration methods. We specifically recognize 3 main types of valuable evaluations: First, robustness evaluation, i.e., consistency of results in the face of data perturbations or alterations in method's parameters. MintTea inherently improves robustness by requiring that modules are steady across multiple data perturbations (e.g., subsampling), and indeed this addition yielded lower false discovery rates as previously noted. A second type of evaluation may concern the generalizability of the obtained findings, i.e., the ability to replicate results in independent cohorts. One limitation of our study is the lack of such cross-study validations of multi-omic modules, which stems from the challenges involved in obtaining unified multi-omic data from different cohorts. We note that as multi-omic data sharing becomes more common and more standardized, such validations will become essential. Lastly, and perhaps most importantly, results should, ideally, be evaluated for their biological coherence and relevance. Indeed, this represents a major caveat of MintTea (and other multi-omic integration methods that are purely statistical in nature). Unfortunately, systematically validating the meaningfulness of inferred interactions between multi-omic features is challenging, largely since systematic mapping of such multi-omic mechanisms is lacking or unknown. Accordingly, we encourage users of MintTea to approach result interpretation with caution.

We also note that clearly, any attempt to identify disease-associated modules based on metagenomic and other features can only be as accurate and reliable as the underlying features. Though there are several well-established tools for taxonomic and functional profiling from metagenomics data, they may each induce biases and inaccuracies[104]. Moreover, additional, possibly invaluable features can be recovered from metagenomics and metabolomics data, for example, by using assembly-based methods to add unknown-genome abundances or strain-level information to taxonomic profiles[104], or adding unannotated metabolites from untargeted metabolomic profiles. Another limitation is the insufficient control for major confounders known to exert microbiome shifts (e.g., stool consistency, diet, and medication usage)[105]. This limitation could be potentially mitigated by careful sample stratification, inclusion of these factors as covariates in models, or even adding such data as an additional "view". Diet, for example, is likely a substantial mediator of many of the identified microbiome-metabolome modules, and it could therefore be especially interesting to include "diet"-view profiles in future analyses to elucidate potential diet-driven modules. Finally, since microbiome omic data has unique characteristics, linear approaches such as CCA may be insufficient, and alternative algorithmic and statistical approaches should definitely be explored and developed.

Overall, this work serves as a proof of concept for the potential benefit of applying advanced intermediate integration methods to microbiome multi-omic data for generating multifaceted hypotheses concerning the role of the microbiome in human disease. We advocate for using such multi-omic intermediate integration models to capture complementary biological insights into microbiome-disease associations.

## Methods
### Dataset collection
In this work, we collected, processed, and analyzed data from a few different multi-omic studies, as well as from the curatedMetagenomicData[94] resource. Below are details describing the data obtained from each of these sources, and further processing we performed. We first obtained data from 3 case-control studies that profiled both stool shotgun metagenomics and metabolomics, including an IBD cohort (Franzosa et al.[12]), a CRC cohort (Yachida et al., 2019[11]), and an end-stage renal disease (ESRD) cohort (Wang et al.[32]). For each of these studies we ran HUMAnN3[106] (version v3.0.1) on the raw metagenomic sequencing files to obtain MetaCyc[107] pathway profiles. MetaPhlAn3[106] (version 3.0.14) was used to obtain taxonomic profiles at the species level. Processed metabolomics data was obtained from the supplementary information of each of the original publications or online repositories. Only identified metabolites (i.e., annotated with metabolite name by the original study's authors) were included in the analysis, to allow downstream interpretation. Metadata per subject was obtained from the supplementary information. When distinct disease subtypes were present, we split the data into separate datasets per disease subtype (or kept only a specific subtype), taking each subtype as the "case" patients, and the same set of control subjects as shared controls. Specifically, the IBD cohort[12] was split into Chron's disease (CD) and ulcerative colitis (UC), and in the CRC cohort[11] we kept only late-stage cancer patients (Supplementary Data File S1). In both of these cases, the original publications indicated unique microbiome signatures for each disease subtype/stage and performed separate analyses accordingly, as we do here.

We additionally included data from the European MetaCardis cohort[13,53], in which participants underwent deep clinical phenotyping combined with gut microbiome and serum metabolome profiling. Pre-processed taxonomic, functional, and serum metabolome profiles were downloaded from zenodo.org/records/6242715. Notably, taxonomic profiles were obtained using the mOTU[108] approach, and functional profiles at the KEGG-module level were generated using a custom pipeline. Full data processing details are described in the original publication[53]. As the MetaCardis cohort is inherently divided into several study groups based on metabolic and cardiac phenotypes, we focused on two specific conditions, T2D and metabolic syndrome (without T2D or coronary artery disease). Subjects with antibiotics indications ('ANTIBIOTICS_TOTAL' > 0) or multiple drugs intake ('DRUGTOTAL' ≥ 3) were excluded. As before, the same set of control samples were used for both comparisons.

Lastly, 14 processed shotgun metagenomics datasets were extracted from the "curatedMetagneomicData" R package[94], version 3.2.3, together with the corresponding sample metadata. Specifically, MetaPhlAn3[106] species-level abundances and HUMAnN3[106] unstratified pathway-level (MetaCyc) abundances were obtained. We only considered datasets with data from at least 40 stool samples from different subjects in each study group (as defined by the "study_condition" variable). Longitudinal studies were excluded. Additional details for each dataset are available in Supplementary Data File S1.

In total, data analyzed in this study included 2,629 samples, obtained from 16 studies, and partitioned into 20 case-control datasets, covering 13 disease states. We preliminary focused on 9 datasets (all multi-omic datasets and 3 additional shotgun-only datasets), and in the last section expand the analysis to all datasets.

### Shared data preprocessing
For each of the datasets described above, we obtained sample metadata, taxonomy abundances ('T'), pathway abundances ('P'), and, when available, fecal/serum metabolite abundances ('M'/'S') (Fig. 1B). For all datasets, T and P profiles were converted into relative abundances (by total sum scaling), metabolite values were log-transformed, and constant and rare features (defined as those with <15% non-zero abundance values, or a mean abundance

<0.005% for taxonomy/pathways) were discarded. Moreover, extremely highly correlated features, which likely represent technical artifacts rather than biological phenomena, were clustered (to avoid misleading interpretation downstream) such that absolute correlations between any two cluster members is higher than 0.99 and using one random feature as cluster representative (Supplementary Data File S8). Lastly, non-bacterial MetaCyc pathways were removed from the P profiles, as annotated in the MetaCyc database within the "Expected Taxonomic Range" field.

### Identifying disease-associated multi-omic modules with "MintTea"

To identify multi-omic sets of features that are both tightly associated with each other and collectively associated with the disease, we developed "MintTea" (Multi-omic INTegration Tool for microbiomE Analysis). MintTea is based on sparse generalized canonical correlation analysis (sGCCA)[36,37] and additional methods that extend sGCCA to a supervised scenario by encoding the disease label of each subject as an additional single-variable omic[44,51,52,109]. Briefly, sGCCA searches for a sparse linear transformation per feature table that yields maximal correlations between the respective latent variables, as well as between these variables and the label (see also Supplementary Note 2). Herein, we define the set of features across all omics that were assigned a non-zero loading by sGCCA, as a "putative module". SGCCA can further find additional linear transformations, orthogonal to the previous ones, by a process called matrix deflation[37]. Each such additional set of omic transformations yields an additional "putative module".

MintTea pipeline, as also illustrated in Fig. 1A, includes the following steps: After data preprocessing described above, MintTea performs a repeated subsampling procedure so that for each subsample of the data (80% of the samples, 100 iterations) sGCCA is applied and the first 5 resulting putative modules are recorded. Next, MintTea identifies "consensus modules", by constructing a network of multi-omic features where 2 features are connected by an edge if they appeared together in a putative module in at least 80% of the iterations. Connected components in the graph of size ≥ 2 are then considered "consensus modules". Lastly, to assess the extent to which each identified consensus module adheres to the desired properties described above, MintTea further evaluates each such consensus module preserving only those that meet certain criteria. Specifically, to evaluate whether a given consensus module is associated with the disease state, MintTea applies PCA on all features included in that module (scaled and centered) and takes the first PC as a single representative of the module. It then computes the area under the receiver operating characteristic curve (AUC) for this first PC of each module, and compares it to "null" modules where the same number of features, from the same omics, are randomly selected. MintTea further evaluates, per module, how well it captures associations between different omics, by calculating the average Spearman correlation between pairs of features from different omics within that module (Supplementary Data File S4). Consensus modules with an AUC > 0.7 and above that of random modules, and an average Spearman correlation between features from different omics that was also above that of random modules, are considered 'disease-associated multi-omic modules'. Importantly, all parameters mentioned above can be adjusted by the user (see below). Supplementary Note 1 further describes a conceptual and quantitative comparison between MintTea and related methods.

The pipeline was developed in R using the packages "mixOmics"[109] (version 6.18.1, "block.splsda" function), and "igraph"[110] (version 1.3.4) for graph operations. Classification performances were generally estimated using repeated 10-fold cross-validation and averaged over folds and repeats.

### MintTea pipeline parameters and sensitivity analysis

The MintTea pipeline has several user-adjustable parameters required for tuning the process to specific datasets. Different parameters may be favored based on the size of the outputted modules and downstream analyses, correlation structures in the data, dimensionality of the data, etc. Briefly, these parameters include: 'keep'−the number of features to select from each omic as provided to sGCCA's computation; 'des'−the value to set as design matrix default (see Supplementary Note 2); 'nrep'−number of repetitions; 'nfol'−number of folds to which the entire data is split in order to run sGCCA on $nfol$-1 folds at a time (i.e., subsampling ratio); 'ncom'−number of orthogonal sGCCA components; 'edge'−the threshold defining how often should a pair of features co-appear in a putative module in order for them to co-appear in a consensus module. Exact parameters used per dataset in our analysis are listed in Supplementary Data File S2. Supplementary Fig. S5 shows an example of a sensitivity analysis applied to a specific module from a specific dataset, assessing that module's sensitivity to different parameter choices. Overall, and as demonstrated in Supplementary Fig. S5, modules are generally robust to choice of parameters.

### Modules overlap analysis

To identify significantly overlapping modules from different datasets, we first narrowed the consensus modules to the set of features available in both datasets, and then ran Fisher's exact test to determine odds ratios and $p$ values. Results are listed in Supplementary Data File S7.

### Predicting disease state using simple, early-integration random forest classifiers

To provide a reference value for the AUC's estimations of MintTea's multi-omic modules, we implemented a standard ML pipeline to assess disease predictability in each dataset. Input data was a simple concatenation of all features from all omics (also known as an "early integration" approach). Training was performed with a repeated 5-fold cross-validation (10 repeats) procedure and included reducing dimensionality with a feature selection step (using Boruta[111]), training a random forest (RF) classifier, evaluating model performance on out-of-fold data using AUC, and evaluating feature importance (as described below).

Final model performances, for each dataset, were determined by averaging AUC over the 5 repeats and 10 folds (i.e. a total of 50 trained models).

We used the 'ranger' R package (version 0.14.1)[112] for RF modeling with default parameters. We opted for RF due to its simplicity on the one hand and its tendency to perform well on microbiome abundance data as consistently shown in previous studies[25,113–117]. Hyper-parameter tuning for the RF classifiers with a nested cross-validation approach did not result in significantly improved performance for most datasets and was therefore discarded. 'Boruta' R package (version 7.0.0)[111] was used for feature selection. All results from RF modeling are provided in Supplementary Data File S5.

### Random forest's feature importance analysis

We examined the features importance in each model that achieved an AUC > 0.7, as follows: Within each fold, we calculated permutation-based feature importance scores implemented in 'ranger' package[112]. We additionally applied the method introduced by Altmann et al.[118] to assign a p-value for each feature in each model. Final feature importance per feature was averaged over repeats and folds. Final p-values were FDR-corrected over all features in each feature set. We refer to features that were selected by the feature selection method in at least 50% of repeats/folds and obtained a final Altmann FDR < 0.1 as "contributors". Feature importance results are provided in Supplementary Data File S6.

**Reporting summary**

Further information on research design is available in the Nature Portfolio Reporting Summary linked to this article.

## Data availability

Data used in this study were retrieved from either the curatedMetagenomicData package[94] (version 3.2.3., available at: https://waldronlab.io/curatedMetagenomicData/), supplementary files and deposited data from 4 specific studies[11,12,32,53] (exact source of each table and accession numbers are listed in Supplementary Data File S1). Functional microbiome profiles were based on metabolic pathways in either the MetaCyc (https://metacyc.org/) or KEGG databases.

## Code availability

The R code used for this analysis, including the MintTea pipeline, is available on GitHub (https://github.com/borenstein-lab/multi_view_integration_analysis), and on Zenodo (DOI: 10.5281/zenodo.10707477).

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

## Acknowledgements

We would like to thank the Borenstein lab members for their helpful feedback and insightful discussions during the process. We thank all authors of the multiple microbiome studies whose data were included in our analysis for generating invaluable data and making them publicly available. We are also grateful to Professor Uri Gophna for his important input, support, and assistance. Lastly, we also thank the Tel-Aviv University Computer Science IT team for their technical support during computational work. This work was supported in part by National Institutes of Health grant U19AG057377, and Israel Science Foundation grant 2435/19 to E.B. E.M. and I.S. are supported in part by a fellowship from the Edmond J. Safra Center for Bioinformatics at Tel-Aviv University. The funders had no role in study design, data collection and analysis, decision to publish, or preparation of the manuscript. The icons used in Fig. 1 were designed by 'Freepik' and are available through the 'Flaticon license' (www.flaticon.com).

## Author contributions

E.M. and E.B. conceived the study, interpreted the findings, and wrote the manuscript. E.M. developed MintTea, implemented the analysis pipelines, and conducted the analysis. I.S. helped implementing an early version of this pipeline. All authors read and approved the final manuscript.

## Competing interests

The authors declare no competing interests.
