## [Peer Review File · Nature Communications]

Multi-omic integration of microbiome data for identifying disease-associated modulesEditorial Note: This manuscript has been previously reviewed at another journal that is not operating a transparent peer review scheme. This document only contains reviewer comments and rebuttal letters for versions considered at *Nature Communications*.

REVIEWERS' COMMENTS

Reviewer #1 (Remarks to the Author):

I appreciate the authors' efforts to address my comments, and I feel that the quality of the manuscript has improved. Although most of my previous comments have been addressed, I still have some major concerns regarding the usefulness of the MintTea approach, and the interpretation of the multi-omics modules identified by MintTea.

The following are my major and minor comments:

Major comments:

1. The manuscript suggests that multi-omics modules identified by MintTea provide a more comprehensive understanding of the underlying microbiome-disease mechanisms. This line of thinking regarding findings from statistical approaches is now too overused in the microbiome literature and rarely translates into useful, actionable insights. Do we really need another correlation-based method to find so-called "signatures" of disease? Also, it's still unclear whether modules identified using the MintTea approach have biological relevance to the diseases. Also, low correlation values (0.2–0.4 in magnitude) between features of a module from different omics data, as shown in Figure 2B, raise concerns about MintTea's ability to capture the interrelated multi-omics features and group them into multi-omics modules.
2. The motivation behind identifying the shared multi-omics modules across datasets of various diseases is unclear. What is this for? Are the authors proposing that multiple diseases have the same inter-omic connections, and what does that really mean? Instead, identifying shared multi-omics modules across datasets of the same diseases would be more interesting from a clinical perspective.
3. Several multi-omics modules identified by MintTea include microbiome-derived metabolic pathways. It is important to note that a module may contain metabolic pathways that come from specific gut microbes that may not be a part of that multi-omics module, which raises concerns about the biological relevance of the identified modules. Therefore, considering the source organisms of the pathways is crucial to ensuring that the modules are meaningful and informative.
4. In a nutshell, MintTea is a concept that uses computational methods on multi-omics data to provide feature sets or modules in the context of microbiome-related diseases. However, the authors have not adequately explained how these disease-associated multi-omics modules are connected to the "mechanism" compared to mere statistical associations.
-- For instance, section Lines 241–242 is titled "Disease-associated multi-omic modules reveal multifaceted biological mechanisms". In molecular biology, the term "mechanism" usually refers to the detailed processes or pathways by which biological phenomena occur. It encompasses the actual molecular, cellular, and physiological events that underlie a specific biological function or phenomenon. However, the authors seem to focus mainly on correlations, which, in the opinion of this reviewer, falls very short of revealing mechanisms.
-- The phrase "mechanisms" should be carefully used/considered/revised across the manuscript.
5. The section titled "Multi-omic modules across diverse cohorts and disease states" mainly discusses the common modules or features present in different datasets or diseases. However, the authors have provided few interpretations in this section, which makes it difficult to understand the purpose of this analysis. What is the purpose of detecting prevalent and recurring multi-omics (mentioned in lines 350–351)? What do these overlapping features mean, and ultimately, what can we learn from this? It is worth noting that the previous version of the manuscript provided some

interpretations and explained the purpose of this analysis before the revision.

-- In addition, since the authors have connected the association between *B. uniformis* and B vitamins (mentioned in lines 370–380), demonstrating the correlation between *B. uniformis* and B vitamins in fecal/serum metabolomics may confirm author's interpretation.

Minor comments:

1. Line 348. Shared modules identified across datasets should not be called "multi-omics" modules since the analysis only utilized metagenomics data.
2. In Figure 1A's sGCCA transformation, what does H -> D mean? My guess is "Y-axis: healthy to disease".
3. In Figure 1C, having color legends for the bar graphs would be appreciated. This reviewer had to guess the colors inherited from Figure 1B.
4. The GitHub is slightly outdated.

Reviewer #1 (Remarks on code availability):

This reviewer did not run into any trouble accessing and testing the authors' code.

Reviewer #2 (Remarks to the Author):

This is a comprehensive revision that has addressed my comments, and I have no further concerns.

Response to reviewers

Below, we provide a point-by-point response to the reviewers' comments. Reviewer comments are given in black, responses in blue.

Responses to reviewer #1

"I appreciate the authors' efforts to address my comments, and I feel that the quality of the manuscript has improved. Although most of my previous comments have been addressed, I still have some major concerns regarding the usefulness of the MintTea approach, and the interpretation of the multi-omics modules identified by MintTea."

We again thank the reviewer for his/her thorough reviewing of the revised manuscript, and for the comments below. See our answers inline to each comment.

"The following are my major and minor comments:

Major comments:

1. The manuscript suggests that multi-omics modules identified by MintTea provide a more comprehensive understanding of the underlying microbiome-disease mechanisms. This line of thinking regarding findings from statistical approaches is now too overused in the microbiome literature and rarely translates into useful, actionable insights. Do we really need another correlation-based method to find so-called "signatures" of disease? Also, it's still unclear whether modules identified using the MintTea approach have biological relevance to the diseases. Also, low correlation values (0.2–0.4 in magnitude) between features of a module from different omics data, as shown in Figure 2B, raise concerns about MintTea's ability to capture the interrelated multi-omics features and group them into multi-omics modules."

We understand and appreciate the reviewer's criticism, yet we want to note the following points as we believe they provide important context:

- First, we argue that even when considering the potential limitations of statistical methods (such as MintTea), searching for microbiome features that change in disease is perhaps the most common theme of data-driven, case-control, studies, and that such statistical methods are still essential for making sense of such studies. Accordingly, development and improvement of statistical methods better tailored for microbiome research is still an active area of research. Our impression from numerous multi-omic studies we have reviewed is that nowadays typical data analyses include differential abundance testing of single features, and correlation networks between the different omics. Accordingly, in the context of current practices, we do believe that MintTea offers an alternative and complementary approach to these analyses by searching for a different type of pattern in the data: small modules that span all omics and capture relationships between omics and with the disease. Furthermore, even though these modules are backed-up by statistics only (as any other statistical method), they can still offer a different type of insight and perspective into the data collected in such studies. For example, given a list of statistically significant differentially abundant taxa, MintTea may reveal that the top species enriched in patients with a disease has correlations, high above those expected at random, with a few other species and with several fecal metabolites. The researcher is thus offered additional exploration directions that in contrast to hypotheses generated by the other mentioned methods, capture all types of data and phenotype associations combined.
- Regarding the relatively low correlation coefficients (most of which are, importantly, statistically significant after FDR), it should be noted that it is in fact not common to observe high correlations between features in metagenomics data, let alone between metagenomic features to other-omic features. Below, as an example, we plot the distribution of correlation

coefficients between fecal taxa and metabolites (on the top), and fecal taxa and serum metabolites (on the bottom), in two different datasets. As you can see, correlations of 0.2-0.4 (absolute values) may indeed be considered noteworthy, depending on the dataset and the omics. We therefore argue that *statistically significant* correlations, even with coefficients of 0.2-0.4, are indeed interesting when examining them in the context of multi-omic data and in the context of the distribution of correlations presented below.

“2. The motivation behind identifying the shared multi-omics modules across datasets of various diseases is unclear. What is this for? Are the authors proposing that multiple diseases have the same inter-omic connections, and what does that really mean? Instead, identifying shared multi-omics modules across datasets of the same diseases would be more interesting from a clinical perspective.”

We believe that a very natural follow up question after detecting multi-omic modules in each dataset, is whether or not these modules tend to overlap across cohorts. We completely agree with the reviewer that identifying shared modules across cohorts of the same disease would have been more interesting clinically. Unfortunately, however, considering the availability of data and discrepancies between studies, we weren’t able to find diseases for which multiple different high-quality datasets with the same omics and relatively similar cohort characteristics were available. Hence, we instead examined shared trends across datasets that MintTea was able to capture, even if they aren’t in the context of a disease. We agree, however, following this comment, that a note to readers on this issue could be beneficial, and so we now mention this point in the discussion (lines 402-406). We also removed the words “across diseases” from the title of this section.

“3. Several multi-omics modules identified by MintTea include microbiome-derived metabolic pathways. It is important to note that a module may contain metabolic pathways that come from specific gut microbes that may not be a part of that multi-omics module, which raises concerns about the biological relevance of the identified modules. Therefore, considering the source organisms of the pathways is crucial to ensuring that the modules are meaningful and informative.”

We agree with the reviewer that ideally, we would have wanted modules to reflect known relationships between metabolic pathways and bacterial taxa (e.g. specificity of a certain pathway to a certain taxon). Examining our modules, we see that often this is indeed the case (and describe such relationships, e.g. in the Cirrhosis module described in lines 322 and on), yet in other modules we weren't always able to explicitly link between taxa and pathways based on online databases or a literature search. The lack of a clear link between taxa and pathways may be attributed to many different factors, including, among others, inaccurate mapping, incomplete annotations, strain-level diversity, co-occurrence patterns, and missing information about metabolic capacities.

Moreover, though an important point, we argue that a systematic validation of biologically meaningful pathway-taxon links is, unfortunately, not in the scope of this study, for two main reasons: First, both our taxonomic and functional profiles are based on commonly-used methods (e.g. MetaPhlAn and HUMAnN) and databases (e.g. NCBI and MetaCyc) that though popular, are not without limitations (as we also note in the discussion, lines 444 and on; See also an entire discussion about the problems with pathway estimation methods for metagenomes here: doi.org/10.1186/s40168-017-0231-4). Accordingly, inaccuracies in pathway/taxon annotations or spurious correlations in the data, may in turn be reflected by MintTea (as for any statistical method) findings. Second, systematic examination of taxon-pathway relationships as part of a computational pipeline is in fact far from being straightforward, as it requires annotations mappable to some common database linking the two entity types, which though hypothetically possible, will also result in biases and inaccuracies (i.e. linking NCBI organisms to KEGG or BioCyc). In the modules we discuss, for example, most of the support we found for metabolic pathways confirmed in specific species was based on literature searches, which is not a process easily automated.

More generally, it should be noted that our lab is in fact an avid advocate for incorporating prior knowledge into computational frameworks (and is often considered a pioneer in the field). However, as the reviewer highlighted in his/her comments above, in this work we introduce a purely statistical method that aims to identify sets of features that correlate with one another higher than expected by chance. allowing researchers to apply this framework to their feature tables, to obtain such statistical signatures, and to interpret or validate them as they find appropriate for their research.

“4. In a nutshell, MintTea is a concept that uses computational methods on multi-omics data to provide feature sets or modules in the context of microbiome-related diseases. However, the authors have not adequately explained how these disease-associated multi-omics modules are connected to the “mechanism” compared to mere statistical associations.

-- For instance, section Lines 241–242 is titled “Disease-associated multi-omic modules reveal multifaceted biological mechanisms”. In molecular biology, the term “mechanism” usually refers to the detailed processes or pathways by which biological phenomena occur. It encompasses the actual molecular, cellular, and physiological events that underlie a specific biological function or phenomenon. However, the authors seem to focus mainly on correlations, which, in the opinion of this reviewer, falls very short of revealing mechanisms.

-- The phrase “mechanisms” should be carefully used/considered/revised across the manuscript.”

We thank the reviewer for his/her important comment and agree that the term ‘mechanism’ may suggest a more detailed and principled explanation to a molecular biologist that we intended in our work.

Following this comment we thus revised the text to remove declarations regarding “mechanisms” (e.g. in lines 226, 464, and others), and underline that MintTea findings are purely statistical (lines 437-443).

“5. The section titled “Multi-omic modules across diverse cohorts and disease states” mainly discusses the common modules or features present in different datasets or diseases. However, the authors have provided few interpretations in this section, which makes it difficult to understand the purpose of this analysis. What is the purpose of detecting prevalent and recurring multi-omics (mentioned in lines 350–351)? What do these overlapping features mean, and ultimately, what can we learn from this? It is worth noting that the previous version of the manuscript provided some interpretations and explained the purpose of this analysis before the revision.

-- In addition, since the authors have connected the association between *B. uniformis* and B vitamins (mentioned in lines 370–380), demonstrating the correlation between *B. uniformis* and B vitamins in fecal/serum metabolomics may confirm author’s interpretation.”

We point the reviewer to our answer to comment #2 above, where we described our general motivation for this analysis and the clarification we have added following these comments. As for the suggestion to directly demonstrate correlations between B vitamins detected through metabolomics and *B. uniformis* abundances, it is unfortunately not possible in the datasets analyzed in the relevant section as they do not include metabolomic data. As an alternative to measuring direct correlations, we thus conducted a literature search and provided several references in the manuscript supporting *B. uniformis* possible role in Vitamin B related metabolism. Moreover, following this comment, we further searched in a previous study from our group (doi: 10.1038/s41522-022-00345-5, Supplementary Table 4), in which we systematically and rigorously analyzed microbiome-metabolome associations across multiple studies, and found that the *Bacteroides* genus was indeed significantly and consistently correlated with Pyridoxamine (a form of vitamin B6), as determined by a meta-analysis conducted on 5 different datasets in which both the genus and the metabolite were detected. Though this finding is at the genus level only, we also know that *B. uniformis*, *B. thetaiotaomicron*, and *B. ovatus* (which are also shown in Figure 3C) are among the most prevalent species in the *Bacteroides* genus in adults (doi: 10.1080/19490976.2020.1848158), thus offering another support to our interpretation of these modules. We have now added this additional support to the manuscript (lines 358-364).

“Minor comments:

6. Line 348. Shared modules identified across datasets should not be called “multi-omics” modules since the analysis only utilized metagenomics data.”

Indeed, in the original manuscript we used the term “multi-view” as we felt it may be more appropriate for cases where we used multiple profiles obtained from the same molecular assay. Nonetheless, a few of the comments we received in the latest revision suggested that the term “multi-view” is less familiar in our domain and should be replaced with “multi-omic” to underscore the main usage scenario intended for our method. Hence, we believe that replacing the general “multi-omic” term in this section alone may be confusing for readers, especially given that the manuscript already includes multiple new terms.

“7. In Figure 1A’s sGCCA transformation, what does H -> D mean? My guess is “Y-axis: healthy to disease”.”

Following this comment, a note on this matter was added to the legend of Figure 1A.

“8. In Figure 1C, having color legends for the bar graphs would be appreciated. This reviewer had to guess the colors inherited from Figure 1B.”

A legend was added to 1C.

“9. The GitHub is slightly outdated.”

March 6, 2024

We couldn't recover any outdated materials or code in the repository. Slight terminology updates in the main README file (https://github.com/borenstein-lab/multi_view_integration_analysis/blob/main/src/intermediate_integration/README.md) were performed on January 16th, and the link to the preprint was updated on January 24th.

Response to reviewer #2

“This is a comprehensive revision that has addressed my comments, and I have no further concerns.”

We sincerely thank the reviewer for taking the time to review our manuscript once again and for his/her support.

Sincerely,

Elhanan Borenstein, Ph.D.

Professor, Blavatnik School of Computer Science, Tel Aviv University

Professor, Faculty of Medicine, Tel Aviv University

Edmond J. Safra Center for Bioinformatics, Tel Aviv University

Affiliate Professor, Department of Genome Sciences, University of Washington

External Professor, Santa Fe Institute